# Robust Option Learning for Adversarial Generalization

## Abstract

Compositional reinforcement learning is a promising approach for training poli-
cies to perform complex long-horizon tasks. Typically, a high-level task is decom-
posed into a sequence of subtasks and a separate policy is trained to perform each
subtask. In this paper, we focus on the problem of training subtask policies in a
way that they can be used to perform any task; here, a task is given by a sequence
of subtasks. We aim to maximize the worst-case performance over all tasks as
opposed to the average-case performance. We formulate the problem as a two
agent zero-sum game in which the adversary picks the sequence of subtasks. We
propose two RL algorithms to solve this game: one is an adaptation of existing
multi-agent RL algorithms to our setting and the other is an asynchronous version
which enables parallel training of subtask policies. We evaluate our approach on
two multi-task environments with continuous states and actions and demonstrate
that our algorithms outperform state-of-the-art baselines.

## 1   Introduction

Reinforcement learning (RL) has proven to be a promising strategy for solving complex control
tasks such as walking [13], autonomous driving [17], and dexterous manipulation [3]. However, a
key challenge facing the deployment of reinforcement learning in real-world tasks is its high sample
complexity—to solve any new task requires a training a new policy designed to solve that task. One
promising solution is *compositional reinforcement learning*, where individual *options* (or *skills*) are
first trained to solve simple tasks; then, these options can be composed together to solve more
complex tasks [25, 24, 17]. For example, if a driving robot learns how to make left and right turns
and to drive in a straight line, it can then drive along any route composed of these primitives.

A key challenge facing compositional reinforcement learning is the generalizability of the learned
options. In particular, options trained under one distribution of tasks may no longer work well if used
in a new task, since the distribution of initial states from which the options are used may shift. An
alternate approach is to train the options separately to perform specific subtasks, but options trained
this way might cause the system to reach states from which future subtasks are hard to perform. One
can overcome this issue by handcrafting rewards to encourage avoiding such states [17], in which
case they generalize well, but this approach relies heavily on human time and expertise.

We propose a novel framework that addresses this challenges by formulating the option learning
problem as an adversarial reinforcement learning problem. At a high level, the adversary chooses
the task that minimizes the reward achieved by composing the available options. Thus, the goal is
to learn a set of *robust options* that perform well across *all* potential tasks. Then, we provide two
algorithms for solving this problem. The first adapts existing ideas for using reinforcement learning
to solve Markov games to our setting. Then, the second shows how to leverage the compositional

Submitted to 36th Conference on Neural Information Processing Systems (NeurIPS 2022). Do not distribute.

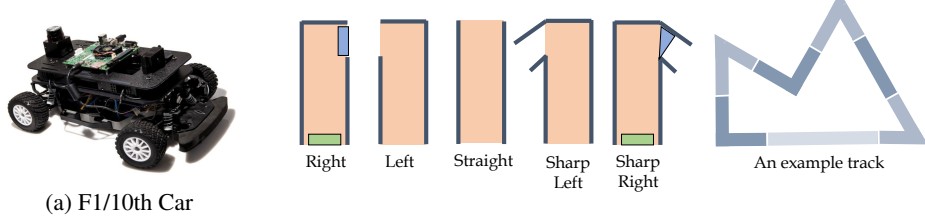

(a) F1/10th Car

(b) Segment Types

Figure 1: F1/10th Environment. The entry and exit regions for the right and sharp right segments are shown in green and blue respectively.

structure of our problem to learn options in parallel at each step of a value iteration procedure; in some cases, by enabling such parallelism, we can reduce the computational cost of learning.

We validate our approach on two benchmarks: (i) a rooms environment where a point mass robot must navigate any given sequence of rooms, where the sequence is an arbitrary combination of straight, left, and right turns, and (ii) a simulated version of the F1/10th car, where a small racing car must navigate any racetrack composed of several different track segments. In both, our empirical results demonstrate that robust options are critical for performing well on a wide variety of tasks.

In summary, our contributions are: (i) a game theoretic formulation of the compositional reinforcement learning problem, (ii) two algorithms for solving this problem, and (iii) an empirical evaluation demonstrating the effectiveness of our approach.

**Motivating example.** Let us consider a small scale autonomous racing car shown in Figure 1 (a). We would like to train a controller that can be used to navigate the car through *all* tracks constructed using five kinds of segments; the possible segments are shown in Figure 1 (b) along with an example track. The state of the car is a vector $(x, y, v, \theta)$ where $(x, y)$ is its position on the track relative to the current segment, $v$ is its current speed and $\theta$ is the heading angle. An action is a pair $(a, \omega) \in \mathbb{R}^2$ where $a$ is the throttle input and $\omega$ is the steering angle. In this environment, completing each segment is considered a subtask and a task corresponds to completing a sequence of segments— e.g., straight $\rightarrow$ right $\rightarrow$ left $\rightarrow$ sharp-right. Upon completion of a subtask, the car enters the next segment and a change-of-coordinates is applied to the car's state which is now relative to the new segment. The goal here is to learn one option for each subtask such that the agent can perform any task using these options.

If one trains the options independently with the only goal of reaching the end of each segment (e.g., using distance-based rewards), it might (and does) happen that the car reaches the end of a segment in a state that was not part of the initial states used to train the policy corresponding to the next subtask. Therefore, one should make sure that the initial state distribution used during training includes such states as well—either manually or using dataset aggregation [38]. Furthermore, it is possible that the car reaches a state in the exit region of a segment from which it is challenging to complete the next subtask—e.g., a state in which the car is close to and facing towards a wall. Our algorithm identifies during training that, in order to perform future subtasks, it is better to reach the end of a segment in a configuration where the car is facing straight relative to the next segment. As demonstrated in our experiments, this leads to robust options and improved sample efficiency.

**Related work.** The options framework [41] is commonly used to model subtask policies as temporally extended actions. In hierarchical RL [32, 31, 22, 9, 5, 43], options are trained along with a high-level controller that chooses the sequence of options to execute in order to complete a specific high-level task. There is also work on discovering options—e.g., using intrinsic motivation [30], entropy maximization [10], semi-supervised RL [12], skill chaining [20], expectation maximization [8] and subgoal identification [40]. There has also been a lot of research on planning using learned options [1, 18, 37, 42, 21].

There has been some work on RL for zero-shot generalization [44, 33, 39, 23, 4]; however, in prior work, the learning objective is to maximize average performance with respect to a fixed distribution over tasks as opposed to the worst-case. Some hierarchical RL algorithms have also been shown to enable few-shot generalization [18] to unseen tasks. Most closely related to our work is the work

on compositional RL in the multi-task setting [17] in which the subtask policies are trained using standard RL algorithms in a naive way without guarantees regarding worst-case performance.

There has also been work on skill composition using transition policies [25]; this method assumes that the subtask policies are fixed and learns one transition policy per subtask which takes the system from an end state to a "favourable" initial state for the subtask. However, poorly trained subtask policies can lead to situations in which it is not possible to achieve such transitions. In contrast, our approach trains subtask policies which compose well without requiring additional transition policies. A recent paper [24] proposes a framework for training subtask policies with the aim of composing them to perform a complex long-horizon task. However, their approach assumes that the high-level task is fixed and the options are trained to maximize the performance with respect to a specific task.

There has been a lot of research on multi-agent RL algorithms [29, 15, 16, 28, 35, 36, 2] including algorithms for two-agent zero-sum games [6, 45, 27]. In this paper, we utilize the specific structure of our game to obtain a simple algorithm that neither requires solving matrix games nor trains a separate policy for the adversary. Furthermore, we show that we can obtain an asynchronous RL algorithm which enables learning options in parallel.

## 2 Problem Formulation

A *multi-task Markov decision process (MDP)* is a tuple $\mathcal{M} = (S, A, P, \Sigma, R, F, T, \gamma, \eta, \sigma_0)$, where $S$ are the states, $A$ are the actions, $P(s' \mid s, a) \in [0, 1]$ is the probability of transitioning from $s$ to $s'$ on action $a$, $\eta$ is the initial state distribution, and $\gamma \in (0, 1)$ is the discount factor. Furthermore, $\Sigma$ is a set of subtasks and for each subtask $\sigma \in \Sigma$, $R_\sigma : S \times A \to \mathbb{R}$ is a reward function[1], $F_\sigma \subseteq S$ is a set of final states where the subtask is considered completed and $T_\sigma : F_\sigma \times S \to [0, 1]$ is the jump probability function; upon reaching a state $s$ in $F_\sigma$ the system jumps to a new state $s'$ with probability $T_\sigma(s' \mid s)$. For the sake of clarity, we assume[2] that $T_\sigma(s' \mid s) = 0$ for any $s'$ with $s' \in F_{\sigma'}$ for some $\sigma'$. Finally, $\sigma_0 \in \Sigma$ is the initial subtask which has to be completed first[3]. A multi-task MDP can be viewed as a discrete time variant of a hybrid automaton model [17].

In the case of our motivating example, the set of subtasks is given by

$$\Sigma = \{\texttt{left}, \texttt{right}, \texttt{straight}, \texttt{sharp-left}, \texttt{sharp-right}\}$$

with $F_\sigma$ denoting the exit region of the segment corresponding to subtask $\sigma$. We use the jump transitions $T$ to model the change-of-coordinates performed upon reaching an exit region. The reward function $R_\sigma$ for a subtask $\sigma$ is given by $R_\sigma(s, a, s') = -\|s' - c_\sigma\|_2^2 + B \cdot \mathbb{1}(s' \in F_\sigma)$ where $c_\sigma$ is the center of the exit region and the subtask completion bonus $B$ is a positive constant.

A task $\tau$ is defined to be an infinite sequence[4] of subtasks $\tau = \sigma_0 \sigma_1 \ldots$, and $\mathcal{T}$ denotes the set of all tasks. For any task $\tau \in \mathcal{T}$, $\tau[i]$ denotes the $i^{\text{th}}$ subtask $\sigma_i$ in $\tau$. In our setting, the task is chosen by the environment nondeterministically. Given a task $\tau$, a configuration of the environment is a pair $(s, i) \in S \times \mathbb{Z}_{\geq 0}$ with $s \notin F_{\tau[i]}$ denoting that the system is in state $s$ and the current subtask is $\tau[i]$. The initial distribution over configurations $\Delta : S \times \mathbb{Z}_{\geq 0} \to [0, 1]$ is given by $\Delta(s, i) = \eta_{\tau[0]}(s)$ if $i = 0$ and $0$ otherwise. The probability of transitioning from $(s, i)$ to $(s', j)$ on an action $a$ is

$$\Pr((s', j) \mid (s, i), a) = \begin{cases} \sum_{s'' \in F_{\tau[i]}} P(s'' \mid s, a) T_{\tau[i]}(s' \mid s'') & \text{if } j = i + 1 \\ P(s' \mid s, a) & \text{if } j = i \\ 0 & \text{otherwise.} \end{cases}$$

Intuitively, the system transitions to the next subtask if the current subtask is completed and stays in the current subtask otherwise. A (deterministic) policy is a function $\pi : S \to A$, where $a = \pi(s)$ is the action to take in state $s$. Our goal is to learn one policy $\pi_\sigma$ for each subtask $\sigma$ such that the discounted reward over the worst-case task $\tau$ is maximized. Formally, given a set of policies $\Pi = \{\pi_\sigma \mid \sigma \in \Sigma\}$ and a task $\tau$, we can define a Markov chain over configurations with initial distribution $\Delta$ and transition probabilities given by $P_\Pi((s', j) \mid (s, i)) = \Pr((s, j') \mid (s, i), \pi_{\tau[i]}(s))$. We denote

---

[1]We can also have $R_\sigma : S \times A \times S \to \mathbb{R}$ depending on the next state but we omit it for clarity of presentation.

[2]This assumption can be removed by adding a fictitious copy of $F_\sigma$ to $S$ for each $\sigma \in \Sigma$.

[3]When there is no fixed initial subtask, we can add a fictitious initial subtask.

[4]A finite sequence can be appended with an infinite sequence of a fictitious subtask with zero reward.

by $\mathcal{D}_{\tau}^{\Pi}$ the distribution over infinite sequences of configurations $\rho = (s_0, i_0)(s_1, i_1)\ldots$ generated by $\tau$ and $\Pi$. Then, we define the objective function as

$$J(\Pi) = \inf_{\tau \in \mathcal{T}} \mathbb{E}_{\rho \sim \mathcal{D}_{\tau}^{\Pi}} \Big[ \sum_{t=0}^{\infty} \gamma^t R_{\tau[i_t]}(s_t, \pi_{\tau[i_t]}(s_t)) \Big].$$

These definitions can be naturally extended to stochastic policies as well. In our motivating example, choosing a large enough completion bonus $B$ guarantees the discounted reward to be higher for runs in which more subtasks are completed. Our aim is to compute a set of policies $\Pi^* \in \arg\max_{\Pi} J(\Pi)$. Each subtask policy $\pi_\sigma$ defines an option [41] $o_\sigma = (\pi_\sigma, I_\sigma, \beta_\sigma)$ where $I_\sigma = S \setminus F_\sigma$ and $\beta_\sigma(s) = \mathbb{1}(s \in F_\sigma)$. Here, the choice of which option to trigger is made by the environment rather than the agent.

## 3 Reduction to Stagewise Markov Games

The problem statement naturally leads to a game theoretic view in which the environment is the adversary. We can formally reduce the problem to a two-agent zero-sum Markov game $\mathcal{G} = (\bar{S}, A_1, A_2, \bar{P}, \bar{R}, \bar{\gamma}, \bar{\eta})$ where $\bar{S} = S \times \Sigma$ is the set of states, $A_1 = A$ are the actions of agent 1 (the agent learning the options) and $A_2 = \Sigma$ are the actions of agent 2 (the adversary). The transition probability function $\bar{P} : \bar{S} \times A_1 \times A_2 \times \bar{S} \to [0, 1]$ is given by

$$\bar{P}((s', \sigma') \mid (s, \sigma), a_1, a_2) = \begin{cases} P(s' \mid s, a_1) & \text{if } s \notin F_\sigma \ \& \ \sigma = \sigma' \\ T_\sigma(s' \mid s) & \text{if } s \in F_\sigma \ \& \ \sigma' = a_2 \\ 0 & \text{otherwise.} \end{cases}$$

We observe that the states are partitioned into two sets $\bar{S} = S_1 \cup S_2$ where $S_1 = \{(s, \sigma) \mid s \notin F_\sigma\}$ is the set of states where agent 1 acts (causing a step in $\mathcal{M}$) and $S_2 = \{(s, \sigma) \mid s \in F_\sigma\}$ is the set of states where agent 2 takes actions (causing a change of subtask); this makes $\mathcal{G}$ a stagewise game. The reward function $\bar{R} : \bar{S} \times A_1 \to \mathbb{R}$ is given by $\bar{R}((s, \sigma), a) = R_\sigma(s, a)$ if $s \notin F_\sigma$ and 0 otherwise. The discount factor depends on the state and is given by $\bar{\gamma}(s, \sigma) = \gamma$ if $s \notin F_\sigma$ and 1 otherwise; this is because a change of subtask does not invoke a step in $\mathcal{M}$. The initial state distribution $\bar{\eta}$ is given by $\bar{\eta}(s, \sigma) = \eta(s)\mathbb{1}(\sigma = \sigma_0)$. A run of the game is a sequence $\bar{\rho} = \bar{s}_0 a_0^1 a_0^2 \bar{s}_1 a_1^1 a_1^2 \ldots$ where $\bar{s}_t \in \bar{S}$ and $a_t^i \in A_i$.

A (deterministic) policy for agent $i$ is a function $\pi_i : \bar{S} \to A_i$. Given policies $\pi_1$ and $\pi_2$ for agents 1 and 2, respectively and a state $\bar{s} \in \bar{S}$ we denote by $\mathcal{D}_{\bar{s}}^{\mathcal{G}}(\pi_1, \pi_2)$ the distribution over runs generated by $\pi_1$ and $\pi_2$ starting at $\bar{s}$. Then, the value of a state $\bar{s}$ is defined by

$$V^{\pi_1, \pi_2}(\bar{s}) = \mathbb{E}_{\bar{\rho} \sim \mathcal{D}_{\bar{s}}^{\mathcal{G}}(\pi_1, \pi_2)} \Big[ \sum_{t=0}^{\infty} \big( \prod_{k=0}^{t-1} \bar{\gamma}(\bar{s}_k) \big) \bar{R}(\bar{s}_t, a_t^1) \Big].$$

We are interested in computing a policy $\pi_1^*$ maximizing

$$J_{\mathcal{G}}(\pi_1) = \mathbb{E}_{\bar{s} \sim \bar{\eta}}[\min_{\pi_2} V^{\pi_1, \pi_2}(\bar{s})].$$

Given a policy $\pi_1$ for agent 1, we can construct a policy $\pi_\sigma$ for any subtask $\sigma$ given by $\pi_\sigma(s) = \pi_1(s, \sigma)$; we denote by $\Pi(\pi_1)$ the set of subtask policies constructed this way. The following theorem connects the objective of the game with our multi-task learning objective; all proofs are in Appendix A.

**Theorem 3.1.** *For any policy $\pi_1$ for agent 1 in $\mathcal{G}$, we have $J(\Pi(\pi_1)) \geq J_{\mathcal{G}}(\pi_1)$.*

Therefore, $J_{\mathcal{G}}(\pi_1)$ is a lower bound on the objective $J(\Pi(\pi_1))$ which we seek to maximize. Now, let us define the optimal value of a state $\bar{s}$ by $V^*(\bar{s}) = \max_{\pi_1} \min_{\pi_2} V^{\pi_1, \pi_2}(\bar{s})$. The following theorem shows that it is possible to construct a policy $\pi_1^*$ that maximizes $J_{\mathcal{G}}(\pi_1)$ from the optimal value function $V^*$.

**Theorem 3.2.** *For any policy $\pi_1^*$ such that for all $(s, \sigma) \in S_1$,*

$$\pi_1^*(s, \sigma) \in \arg\max_{a \in A} \Big\{ \bar{R}((s, \sigma), a) + \gamma \cdot \sum_{s' \in S} P(s' \mid s, a)V^*(s', \sigma) \Big\},$$

*we have that $\pi_1^* \in \arg\max_{\pi_1} J_{\mathcal{G}}(\pi_1)$.*

---

**Algorithm 1** Asynchronous value iteration algorithm for computing optimal subtask policies.

1: **function** ASYNCVALUEITERATION($\mathcal{M}, V$)
2:     **while** stopping criterion is met **do**
3:         **for** $\sigma \in \Sigma$ **do**     // in parallel
4:             Compute $\mathcal{W}_\sigma(V)$
5:         $V \leftarrow \mathcal{F}_{\texttt{async}}(V)$     // using Equation 3

---

## 3.1 Value Iteration

In this section, we briefly look at two value iteration algorithms to compute $V^*$ which we later adapt in Section 4 to obtain learning algorithms. Let $\mathcal{V} = \{V : S_1 \to \mathbb{R}\}$ denote the set of all value functions over $S_1$. Given a value function $V \in \mathcal{V}$ we define its extension to all of $\bar{S}$ using

$$[\![V]\!](s, \sigma) = \begin{cases} \min_{\sigma' \in \Sigma} \sum_{s' \in S} T_\sigma(s' \mid s) V(s', \sigma') & \text{if } s \in F_\sigma \\ V(s, \sigma) & \text{otherwise.} \end{cases} \tag{1}$$

For a state $s \in F_\sigma$, $[\![V]\!](s, \sigma)$ denotes the worst-case value (according to $V$) with respect to the possible choices of next subtask $\sigma'$. Now, we consider the Bellman operator $\mathcal{F} : \mathcal{V} \to \mathcal{V}$ defined by

$$\mathcal{F}(V)(s, \sigma) = \max_{a \in A} \left\{ \bar{R}((s, \sigma), a) + \gamma \cdot \sum_{s' \in S} P(s' \mid s, a) [\![V]\!](s', \sigma) \right\} \tag{2}$$

for all $(s, \sigma) \in S_1$. Let us denote by $V^* \downarrow_{S_1}$ the restriction of $V^*$ to $S_1$. The following lemma follows straightforwardly giving us our first value iteration procedure.

**Theorem 3.3.** *$\mathcal{F}$ is a contraction mapping with respect to the $\ell_\infty$-norm on $\mathcal{V}$ and $V^* \downarrow_{S_1}$ is the unique fixed point of $\mathcal{F}$ with $\lim_{n \to \infty} \mathcal{F}^n(V) = V^* \downarrow_{S_1}$ for all $V \in \mathcal{V}$.*

Next we consider an *asynchronous* value iteration procedure which allows us to parallelize computing subtask policies for different subtasks. Given a subtask $\sigma$ and a value function $V \in \mathcal{V}$, we define a *subtask MDP* $\mathcal{M}_\sigma^V$ which behaves like $\mathcal{M}$ until reaching a final state $s \in F_\sigma$ after which it transitions to a dead state $\perp$ achieving a reward of $[\![V]\!](s, \sigma)$. Formally, $\mathcal{M}_\sigma^V = (S_\sigma, A, P_\sigma, R_\sigma^V, \gamma)$ where $S_\sigma = S \sqcup \{\perp\}$ with $\perp$ being a special dead state, $P_\sigma(s' \mid s, a) = P(s' \mid s, a)$ if $\perp \neq s \notin F_\sigma$ and $P_\sigma(s' \mid s, a) = \mathbb{1}(s' = \perp)$ otherwise. The reward function is given by $R_\sigma^V(s, a) = R_\sigma(s, a)$ if $\perp \neq s \notin F_\sigma$, $R_\sigma^V(s, a) = [\![V]\!](s, \sigma)$ if $\perp \neq s \in F_\sigma$ and is 0 otherwise. We denote by $\mathcal{W}_\sigma(V)$ the optimal value function of the MDP $\mathcal{M}_\sigma^V$. We then define the asynchronous value iteration operator $\mathcal{F}_{\texttt{async}} : \mathcal{V} \to \mathcal{V}$ using

$$\mathcal{F}_{\texttt{async}}(V)(s, \sigma) = \mathcal{W}_\sigma(V)(s). \tag{3}$$

We can show that repeated application of $\mathcal{F}_{\texttt{async}}$ leads to the optimal value function $V^*$ of the $\mathcal{G}$.

**Theorem 3.4.** *For any $V \in \mathcal{V}$, $\lim_{n \to \infty} \mathcal{F}_{async}^n(V) \to V^* \downarrow_{S_1}$.*

Since each $\mathcal{W}_\sigma(V)$ can be computed independently, we can parallelize the computation of $\mathcal{F}_{\texttt{async}}$ giving us the algorithm in Algorithm 1. We can also show that it is not necessary to compute $\mathcal{W}_\sigma(V)$ exactly. Let $\mathcal{V}_\sigma = \{\bar{V} : S_\sigma \to \mathbb{R}\}$ be the set of all value functions over $S_\sigma$. For a fixed $V \in \mathcal{V}$, let $\mathcal{F}_{\sigma,V} : \mathcal{V}_\sigma \to \mathcal{V}_\sigma$ denote the usual Bellman operator for the MDP $\mathcal{M}_\sigma^V$ given by

$$\mathcal{F}_{\sigma,V}(\bar{V})(s) = \max_{a \in A} \left\{ R_\sigma^V(s, a) + \gamma \cdot \sum_{s' \in S_\sigma} P_\sigma(s' \mid s, a) \bar{V}(s') \right\}$$

for all $\bar{V} \in \mathcal{V}_\sigma$ and $s \in S_\sigma$. For any $V \in \mathcal{V}$ and $\sigma \in \Sigma$, we define a corresponding $V_\sigma \in \mathcal{V}_\sigma$ using $V_\sigma(s) = [\![V]\!](s, \sigma)$ if $s \in S$ and $V_\sigma(\perp) = 0$. Then, for any integer $m > 0$ and $V \in \mathcal{V}$, we can use $\mathcal{F}_{\sigma,V}^m(V_\sigma)$ as an approximation to $\mathcal{W}_\sigma(V)$. Let us define $\mathcal{F}_m : \mathcal{V} \to \mathcal{V}$ using

$$\mathcal{F}_m(V)(s, \sigma) = \mathcal{F}_{\sigma,V}^m(V_\sigma)(s).$$

Intuitively, $\mathcal{F}_m$ corresponds to performing $m$ steps of value iteration in each subtask MDP $\mathcal{M}_\sigma^V$ (which can be parallelized) starting at $V_\sigma$. The following theorem guarantees convergence when using $\mathcal{F}_m$ instead of $\mathcal{F}_{\texttt{async}}$.

**Theorem 3.5.** *For any $V \in \mathcal{V}$ and $m > 0$, $\lim_{n \to \infty} \mathcal{F}_m^n(V) \to V^* \downarrow_{S_1}$.*

**Algorithm 2** Robust Option Soft Actor Critic.
Inputs: Learning rates $\alpha_\psi$, $\alpha_\theta$, entropy weight $\beta$ and Polyak rate $\delta$.

```
 1: function ROSAC(αψ, αθ, β, δ)
 2:     Initialize parameters {ψσ}σ∈Σ, {ψσ^targ}σ∈Σ and {θσ}σ∈Σ
 3:     Initialize replay buffer B
 4:     for each iteration do
 5:         for each episode do
 6:             s0 ∼ η
 7:             σ0 ← InitialSubtask
 8:             for each step t do
 9:                 at ∼ πθσt(· | st) and st+1 ∼ P(· | s, a)
10:                 B ← B ∪ {(st, at, st+1)}
11:                 if st+1 ∈ Fσt then
12:                     st+1 ∼ Tσt(· | st+1)
13:                     σt+1 ← Greedy(ε, arg minσ Ṽ(st+1, σ), Σ)
14:                 else
15:                     σt+1 ← σt
16:         for each gradient step do
17:             Sample batch B ∼ B
18:             for σ ∈ Σ do
19:                 ψσ ← ψσ − αψ∇ψσ LQ(ψσ, B)
20:                 θσ ← θσ − αθ∇θσ Lπ(θσ, B)
21:                 ψσ^targ ← δψσ + (1 − δ)ψσ^targ
```

## 4 Learning Algorithms

In this section, we present RL algorithms for solving the game $\mathcal{G}$. We first consider the finite MDP setting for which we can obtain a modified $Q$-learning algorithm with a convergence guarantee. We then present two algorithms based on Soft Actor Critic (SAC) for the continuous state setting.

### 4.1 Finite MDP

Assuming finite states and actions, we can obtain a $Q$-learning variant for solving $\mathcal{G}$ which we call *Robust Option Q-learning*. We assume that jump transitions $T$ are known to the learner; this is usually the case since jump transitions are used to model subtask transitions and change-of-coordinates within the controller. However, we believe that the algorithm can be easily extended to the scenario where $T$ is unknown.

We maintain a function $Q : S_1 \times A \to \mathbb{R}$ with $Q(s, \sigma, a)$ denoting $Q((s, \sigma), a)$. The corresponding value function $V_Q$ is defined using $V_Q(s, \sigma) = \max_{a \in A} Q(s, \sigma, a)$ and is extended to all of $\bar{S}$ as $[\![V_Q]\!]$. Note that, given a $Q$-function, the extended value function $[\![V_Q]\!]$ can be computed exactly. Robust Option $Q$-learning is an iterative process—in each iteration $t$, it takes a step $((s, \sigma), a_1, a_2, (s', \sigma))$ in $\mathcal{G}$ with $(s, \sigma) \in S_1$ and updates the $Q$-function using

$$Q_{t+1}(s, \sigma, a_1) \leftarrow (1 - \alpha_t)Q_t(s, \sigma, a_1) + \alpha_t(\bar{R}((s, \sigma), a_1) + \gamma[\![V_{Q_t}]\!](s', \sigma)). \tag{4}$$

where $Q_t$ is the $Q$-function in iteration $t$ and $[\![V_{Q_t}]\!]$ is the corresponding extended value function.

Under standard assumptions on the learning rates $\alpha_t$, similar to $Q$-learning, we can show that Robust Option $Q$-learning converges to the optimal $Q$-function almost surely. Here, the optimal $Q$-function is defined by $Q^*(s, \sigma, a) = \bar{R}((s, \sigma), a) + \gamma \sum_{s' \in S} P(s' \mid s, a)V^*(s', \sigma)$ for all $(s, \sigma) \in S_1$. Let $\alpha_t(s, \sigma, a)$ denote the learning rate used in iteration $t$ if $Q_t(s, \sigma, a)$ is updated and 0 otherwise. Then, we have the following theorem.

**Theorem 4.1.** *If $\sum_t \alpha_t(s, \sigma, a) = \infty$ and $\sum_t \alpha_t^2(s, \sigma, a) < \infty$ for all $(s, \sigma) \in S_1$ and $a \in A$, then $\lim_{t \to \infty} Q_t = Q^*$ with probability 1.*

## 4.2 Continuous States and Actions

In the case of continuous states and actions, we can adapt any $Q$-function based RL algorithm such as Deep Deterministic Policy Gradients (DDPG) [26] or Soft Actor Critic (SAC) [14] to our setting. Here we present an SAC based algorithm that we call Robust Option SAC (ROSAC) which is outlined in Algorithm 2. This algorithm, like SAC, adds an entropy bonus to the reward function to improve exploration.

We maintain two $Q$-functions for each subtask $\sigma$, $Q_{\psi_\sigma} : S \to \mathbb{R}$ parameterized by $\psi_\sigma$ and a target function $Q_{\psi_\sigma^{\texttt{targ}}}$ parameterized by $\psi_\sigma^{\texttt{targ}}$. We also maintain a stochastic subtask policy $\pi_{\theta_\sigma} : S \to \mathcal{D}(A)$ for each subtask $\sigma$ where $\mathcal{D}(A)$ denotes the set of distributions over $A$. Given a step $(s, a, s')$ in $\mathcal{M}$ and a subtask $\sigma$ with $s \notin F_\sigma$, we define the target value by

$$y_\sigma(s, a, s') = R_\sigma(s, a) + \gamma [\![V]\!](s', \sigma)$$

where the value $[\![V]\!](s', \sigma)$ is estimated using $\tilde{V}(s', \sigma) = Q_{\psi_\sigma^{\texttt{targ}}}(s', \tilde{a}) - \beta \log \pi_{\theta_\sigma}(\tilde{a} \mid s')$ with $\tilde{a} \sim \pi_{\theta_\sigma}(\cdot \mid s')$ if $s' \notin F_\sigma$. If $s' \in F_\sigma$, we estimate $[\![V]\!](s', \sigma)$ using $\tilde{V}(s', \sigma) = \min_{\sigma' \in \Sigma} \tilde{V}(s'', \sigma')$ where $\tilde{V}(s'', \sigma') = Q_{\psi_{\sigma'}^{\texttt{targ}}}(s'', \tilde{a}) - \beta \log \pi_{\theta_{\sigma'}}(\tilde{a} \mid s'')$ with $\tilde{a} \sim \pi_{\theta_{\sigma'}}(\cdot \mid s'')$ and $s'' \sim T_\sigma(\cdot \mid s')$. Now, given a batch $B = \{(s, a, s')\}$ of steps in $\mathcal{M}$ we update $\psi_\sigma$ using one step of gradient descent corresponding to the loss

$$\mathcal{L}_Q(\psi_\sigma, B) = \frac{1}{|B|} \sum_{(s,a,s') \in B} (Q_{\psi_\sigma}(s, a) - y_\sigma(s, a, s'))^2$$

and the subtask policy $\pi_{\theta_\sigma}$ is updated using the loss

$$\mathcal{L}_\pi(\theta_\sigma, B) = -\frac{1}{|B|} \sum_{(s,a,s') \in B} \mathbb{E}_{\tilde{a} \sim \pi_{\theta_\sigma}(\cdot \mid s)} \big[ Q_{\psi_\sigma}(s, \tilde{a}) - \beta \log \pi_{\theta_\sigma}(\tilde{a} \mid s) \big].$$

The gradient $\nabla_{\theta_\sigma} \mathcal{L}_\pi(\theta_\sigma, B)$ can be estimated using the reparametrization trick if $\pi_{\theta_\sigma}(\cdot \mid s)$ is a Gaussian distribution whose parameters are differentiable w.r.t. $\theta_\sigma$. We use Polyak averaging to update the target $Q$-networks $\{Q_{\psi_\sigma^{\texttt{targ}}} \mid \sigma \in \Sigma\}$.

Note that we do not train a separate policy for the adversary. During exploration, we use the $\varepsilon$-greedy strategy to select subtasks. We first estimate the "worst" subtask for a state $s$ using $\tilde{\sigma} = \arg\min_\sigma \tilde{V}(s, \sigma)$ where $\tilde{V}(s, \sigma)$ is estimated as before. Then the function $\texttt{Greedy}(\varepsilon, \tilde{\sigma}, \Sigma)$ chooses $\tilde{\sigma}$ with probability $1 - \varepsilon$ and picks a subtask uniformly at random from $\Sigma$ with probability $\varepsilon$.

**Asynchronous ROSAC.** We can also obtain an asynchronous version of the above algorithm which lets us train subtask policies in parallel. Asynchronous Robust Option SAC (AROSAC) is outlined in Algorithm 3. Here we use one replay buffer for each subtask. We maintain an initial state distribution $\tilde{\eta}$ over $S$ to be used for training every subtask policy $\{\pi_\sigma\}_{\sigma \in \Sigma}$. $\tilde{\eta}$ is represented using a finite set of states $D$ from which a state is sampled uniformly at random. The value function $\tilde{V} : S \times \Sigma \to \mathbb{R}$ is estimated as before. To be specific, in each iteration, an estimate of any value $\tilde{V}(s, \sigma)$ is obtained on the fly using the $Q$-functions and the subtask policies from the previous iteration.

The SAC subroutine runs the standard Soft Actor Critic algorithm for $N$ iterations on the subtask MDP $\mathcal{M}_\sigma^{\tilde{V}}$ (defined in Section 3)[5] with initial state distribution $\tilde{\eta}$ (defaults to $\eta$ if $D = \emptyset$). It returns the updated parameters along with states $X_\sigma$ visited during exploration with $X_\sigma \subseteq F_\sigma$. The states in $X_\sigma$ are used to update the initial state distribution for the next iteration following the Dataset Aggregation principle [38].

# 5 Experiments

We evaluate our algorithms ROSAC and AROSAC on two multi-task environments; a rooms environment and an F1/10th racing car environment [11].

---

[5]Note that it is possible to obtain samples from $\mathcal{M}_\sigma^{\tilde{V}}$ as long can one can obtain samples from $\mathcal{M}$ and membership in $F_\sigma$ can be decided.

**Algorithm 3** Asynchronous Robust Option Soft Actor Critic.
Inputs: Learning rates $\alpha$, entropy weight $\beta$, Polyak rate $\delta$ and number of SAC iterations $N$.

1: **function** AROSAC($\alpha, \beta, \delta, N$)
2:     Initialize parameters $\Psi = \{\psi_\sigma\}_{\sigma\in\Sigma}$, $\Psi^{\texttt{targ}} = \{\psi_\sigma^{\texttt{targ}}\}_{\sigma\in\Sigma}$ and $\Theta = \{\theta_\sigma\}_{\sigma\in\Sigma}$
3:     Initialize replay buffers $\{\mathcal{B}_\sigma\}_{\sigma\in\Sigma}$ and Initialize $D = \{\}$
4:     **for** each iteration **do**
5:         $\tilde{V} \leftarrow$ OBTAINVALUEESTIMATOR($\Psi, \Theta$)
6:         **for** $\sigma \in \Sigma$ **do**    // in parallel
7:             $\psi_\sigma, \psi_\sigma^{\texttt{targ}}, \theta_\sigma, X_\sigma \leftarrow$ SAC($\mathcal{M}_\sigma^{\tilde{V}}, D, \psi_\sigma, \psi_\sigma^{\texttt{targ}}, \theta_\sigma, \alpha, \beta, \delta, N$)
8:         **for** $\sigma \in \Sigma$ **do**
9:             **for** $s \in X_\sigma$ **do**
10:                 $s' \sim T_\sigma(\cdot \mid s)$ and $D \leftarrow D \cup \{s'\}$

**Rooms environment.** We consider the environment shown in Figure 2 which depicts a room with walls and exits. Initially the robot is placed in the green triangle. The L-shaped obstacles indicate walls within the room that the robot cannot cross. A state of the system is a position $(x, y) \in \mathbb{R}^2$ and an action is a pair $(v, \theta)$ where $v$ is the speed and $\theta$ is the heading angle to follow during the next time-step. There are three exits: left (blue), right (yellow) and up (grey) reaching each of which is a subtask. Upon reaching an exit, the robot enters another identical room where the exit is identified (via change-of-coordinates) with the bottom entry region of the current room. A task is a sequence of directions—e.g., `left` $\rightarrow$ `right` $\rightarrow$ `up` $\rightarrow$ `right` indicating that the robot should reach the left exit followed by the right exit in the subsequent room and so on. Although the dynamics are simple, the obstacles make learning challenging in the adversarial setting.

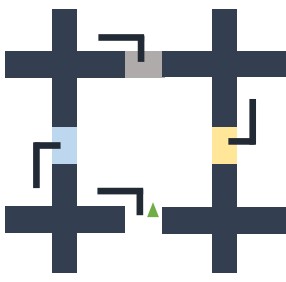

Figure 2: Rooms environment

**F1/10th environment.** This is the environment in the motivating example. A publicly available simulator [11] of the F1/10th car is used for training and testing. The policies use the LiDAR measurements from the car as input (as opposed to the state) and we assume that the controller can detect the completion of each segment; as shown in prior work [17], one can train a separate neural network to predict subtask completion.

**Baselines.** We compare our approach to three baselines. The baseline NAIVE trains one controller for each subtask with the only aim of completing the subtask, similar to [17], using a manually designed initial state distribution. DAGGER is a similar approach which, instead of using a manually designed initial state distribution for training, infers the initial state distribution using the Dataset Aggregation principle [38]. The MADDPG baseline solves the game $\mathcal{G}$ using the multi-agent RL algorithm proposed in [29] for solving concurrent Markov games with continuous states and actions.

**Evaluation.** We evaluate the performance of these algorithms against two adversaries. One adversary is the random adversary which picks the next subtask uniformly at random from the set of all subtasks. The other adversary estimates the worst sequence of subtasks for a given set of options using Monte Carlo Tree Search (MCTS) [19]. The MCTS adversary is trained by assigning a reward of 1 if it selects a subtask which the corresponding policy is unable to complete within a fixed time budget and a reward of 0 otherwise. For the Rooms environment, we consider subtask sequences of length atmost 5 whereas for the F1/10th environment, we consider sequences of subtasks of length at most 20. We evaluate both the average number of subtasks completed as well as the probability of completing the set maximum number of subtasks.

**Results.** The plots for the rooms environment are shown in Figure 3 and plots for the F1/10th environment are shown in Figure 4. We can observe that ROSAC is able outperform other approaches and learn robust options. In the rooms environment, AROSAC achieves similar performace albeit requiring more samples; however, it has the added benefit of being parallelizable. In the F1/10th environment, it performs similar to the other baselines. DAGGER and NAIVE baselines are unable to learn policies that can be used to perform long sequences of subtasks; this is mostly due to the fact

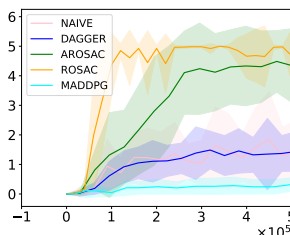 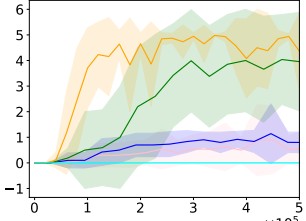 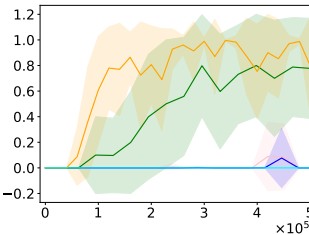

(a) Number of subtaks completed against random adversary

(b) Number of subtasks completed against MCTS adversary

(c) Success probability against MCTS adversary

Figure 3: Plots for the Rooms environment. $x$-axis denoted the number of sample steps and $y$-axis denoted the either the average number of subtasks completed or the probability of completing 5 subtasks. Results are averaged over 10 runs. Error bars indicate $\pm$ standard deviation.

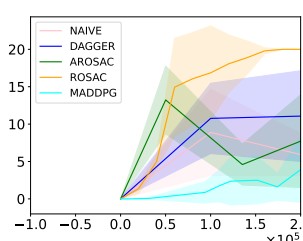 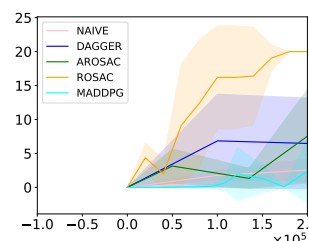 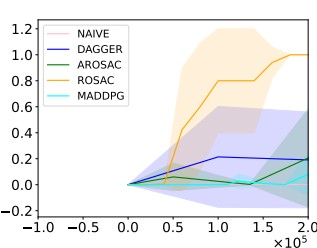

(a) Number of subtaks completed against random adversary

(b) Number of subtasks completed against MCTS adversary

(c) Success probability against MCTS adversary

Figure 4: Plots for the F1/10th environment. $x$-axis denoted the number of sample steps and $y$-axis denoted the either the average number of subtasks completed or the probability of completing 20 subtasks. Results are averaged over 5 runs. Error bars indicate $\pm$ standard deviation.

that they learn options that cause the system to reach states from which future subtasks are difficult to perform—e.g., in the rooms environment, the agent sometimes reaches the left half of the exits from where it is difficult to reach the right exit in the subsequent room. Although MADDPG uses the same reduction to two-player games as ROSAC, it ignores all the structure in the game and treats it as a generic Markov game. As a result, it learns a separate NN policy for each player which leads to the issue of unstable training, primarily due to the non-stationary nature of the environment observed by either agent. As shown in the plots, this leads to poor performance when applied to the problem of learning robust options.

## 6 Conclusions

We have proposed a framework for training robust options which can be used to perform arbitrary sequences of subtasks. In our framework, we first reduce the problem to a two-agent zero-sum stagewise Markov game which has a unique structure. We utilized this structure to design two algorithms, namely ROSAC and AROSAC, and demonstrated that they outperform existing approaches for training options with respect to multi-task performance. One potential limitation of our approach is that the set of subtasks is fixed and has to be provided by the user. An interesting direction for future work is to address this limitation by combining our approach with option discovery methods.

**Societal impacts.** Our work seeks to improve reinforcement learning for complex long-horizon tasks. Any progress in this direction would enable robotics applications both with positive impact—e.g., flexible and general-purpose manufacturing robotics, robots for achieving agricultural tasks, and robots that can be used to perform household chores—and with negative or controversial impact—e.g., military applications. These issues are inherent in all work seeking to improve the abilities of robots.

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
