# OpenReview forum: "Robust Option Learning for Adversarial Generalization"
_NeurIPS.cc/2022/Conference — NeurIPS 2022 Submitted_

### Official Review · Reviewer_KUYx · 2022-07-09

**Rating:** 3
**Confidence:** 3
**Soundness:** 3 good
**Presentation:** 2 fair
**Contribution:** 2 fair

**Summary:**

The paper proposes an approach to learning option policies such that they can be composed sequentially, regardless of the next subtask/option to be performed. The main idea is to reframe the problem as a two-player game, where an adversarial environment arranges the order of the subtasks, and the agent learns one policy for each subtask to maximise the sequential completion of as many as possible in the worst case. The convergence of value iteration to an optimal policy in this setting is proven, while two algorithms based on Q-learning and soft actor-critic are provided. Results in two continuous state/action domains demonstrate that the proposed algorithms significantly outperform the baselines in practice.

**Questions:**

1. The main aim of this paper is to learn robust option policies. As I mentioned in the previous section, much of the difficulty in option discovery is in identifying the subgoals options should reach in the first place. The formulation here sidesteps this problem by simply specifying what those should be. Nonetheless, my larger concern is how this fits into the existing literature. In particular, the line of work revolving around skill chaining seems to tackle the exact problem here: learn options such that they can be sequentially chained together reliably. These works go further though, in that they also do not assume that the subtasks are given. Quoting from [2]: "The chainability of the discovered skills, i.e, the property that successful execution of one permits the execution of another, allows the agent to build a graph that is suitable for deterministic planning". If my understanding is correct, then that line of work is tackling the same problem here and more. In that case, it's unclear how this paper improves upon that line of work. One argument potentially is that these options are reusable, but that is only because of the change of coordinates/relativisation of state variables, which is defined by the system itself. I'm thus unclear how this work advances over the skill chaining work, but I would be interested to hear the authors' thoughts on this. Even if there is a difference, I think that one of the skill chaining approaches should certainly be implemented as a baseline comparison.

2. On the results, one thing I found very confusing is the performance of everything when the adversary is MCTS or random. It seems like a random adversary does just as well as MCTS, so does that not suggest that there is no need to phrase the problem as a 2-player game? The random adversary doesn't seek to minimise anything, so why frame it as a game if it does just as well as one that does? Why not simply always randomise over the subtask order and have the agent train on the randomised sequences?

3. The formulation of the multi-task MDP relies on jump probabilities, change of coordinates, etc, which I've never encountered prior. The paper cites [17], but I'm curious to understand whether this is a common formulation of such problems? It seems to be injecting an awful lot of knowledge and domain expertise into the system and essentially provides the agent with transferrable skills.

4. I found the following sentence in the abstract confusing: "We aim to maximize the worst-case performance..." This put me in mind of other work [4-7] where the aim is to learn a policy or set of policies that are useful for subsequent tasks drawn from some distribution (as opposed to a given set of tasks that must be sequentially executed). I would recommend this be reworded.

5. What does the shading on the results graphs represent? Is it variance?

[1] Bagaria, Akhil, and George Konidaris. "Option discovery using deep skill chaining." International Conference on Learning Representations. 2019.

[2] Bagaria, Akhil, Jason K. Senthil, and George Konidaris. "Skill discovery for exploration and planning using deep skill graphs." International Conference on Machine Learning. PMLR, 2021.

[3] Bagaria, Akhil, et al. "Robustly Learning Composable Options in Deep Reinforcement Learning." IJCAI. 2021.

[4] Alver, Safa, and Doina Precup. "Constructing a Good Behavior Basis for Transfer using Generalized Policy Updates." International Conference on Learning Representations. 2022.

[5] Nemecek, Mark W., and Ron Parr. "Policy caches with successor features." International Conference on Machine Learning. PMLR, 2021.

[6] Jinnai, Yuu, et al. "Discovering options for exploration by minimizing cover time." International Conference on Machine Learning. PMLR, 2019.

[7] Abel, David, et al. "Policy and value transfer in lifelong reinforcement learning." International Conference on Machine Learning. PMLR, 2018.

**Strengths And Weaknesses:**

\+ The two-player formulation is, to my knowledge, a novel way of analysing the problem

\+ Practical algorithms are backed by theoretical guarantees, while empirically the proposed algorithms vastly outperform the given baselines

\+ The introduction of the racing car domain early on really helped to ground the ideas, making the paper much easier to understand

\- Much of the "heavy lifting" is encoded into the system. For example, much of the challenge of option discovery is simply waved away here by pre-specifying the subtasks, reducing the problem to only learning good policies. This is a much easier problem than many other approaches are concerned with (see comments in next section), which makes the contribution fairly incremental.

\- It is unclear how this work fits in with other option-discovery approaches and there are potentially more appropriate baselines to compare against (see next section)

\- I found parts of the paper difficult to read because of the dense notation. In particular, Section 3.1 was slow going, particularly when reading the part about the asynchronous value iteration algorithm.

---

> ### Author Response · Authors · 2022-08-02
> **Response to Reviewer KUYx**
>
> Thanks for helpful comments and pointers to related literature. We address the main concerns below and will incorporate the discussion in the paper.
>
> __Question 1.__ One important observation regarding our approach is that it still has to learn which parts of the goal region to reach for better future performance and only the high level (coarse) goal region is provided to the agent. Therefore, our formalism does not completely sidestep the issue of which subgoal to reach since we only provide high level goal regions which are, in fact, needed to define subtasks.
>
> _Comparison to [1,2]_
> 1. Our aim is to learn options for zero-shot generalization, i.e., the ability to perform subtasks in any sequence without having to retrain the options. In contrast, the option discovery methods [1,2] aim to either discover options for achieving a specific task [1] (reach a pre-specified goal) or discover options that enable reaching various parts of the state space [2].
> 1. In these methods, the policies themselves are either trained using model-free or model-based RL in a similar way to our NAIVE baseline. We believe that our approach is complementary to option discovery methods in the sense that our algorithm can be used to learn the policies corresponding to the options within a skill graph, which would guarantee that any path is achievable with high probability.
>
> _Comparison to [3]_
> 1. [3] tries to learn robust options so that they can be composed, however, they still consider a single overall task and do not consider zero-shot generalization whereas we target worst-case performance across multiple tasks. For example, if a specific track for the F1/10th car is given, then we can apply the approach of [3] whereas we want the options to enable driving in all tracks of a certain kind (there are infinitely many such tracks) and we believe our approach generalizes [3] to this setting in a theoretically grounded and principled way.
> 1. We provide theoretical guarantees regarding the worst-case performance whereas, their approach at best provides a hierarchical optimality guarantee w.r.t. the single task considered.
>
> __Question 2.__ Firstly, on close examination, the performance w.r.t. MCTS adversary is lower for all baselines when compared to their performance against the random adversary. Second, even when measuring against a random adversary, as the plots show, ROSAC outperforms the other approaches indicating our game formalism leads to an algorithm that also improves average-case performance. Therefore, we are not necessarily sacrificing average performance over tasks in order to improve worst-case performance.
>
> __Question 3.__ The notion of jump transitions is commonly used in hybrid automata models for control systems and are derived from there. These jump transitions, however, are also treated similarly to environment transitions and are unknown to the learner (there are parts of the theory where this is assumed known, but these assumptions are easy to remove and were introduced to simplify the math). The agent simply observes completion of a task and what the new state after a jump is. Furthermore, this does not “essentially provide us with transferable skills” since the agent still has to learn the policies and also learn them in a way that they compose well for zero-shot generalization. This is illustrated by the fact that baselines such as NAIVE and DAGGER do not perform well despite being trained on the same environment model and having the same information as our approach.
>
> __Question 4.__ The works [5-7] target multi task generalization either via retraining the policies for new tasks or measure the performance w.r.t. a distribution of tasks as opposed to the worst-case task. [4] considers zero-shot generalization but the tasks considered have to be defined as reward functions which are expressible as linear combinations of a fixed set of features. In contrast, we consider temporal compositions of subtasks to define the set of all tasks which enables us to tackle long horizon problems as opposed to simple tasks considered in [4].
>
> __Question 5.__ Yes, the shading/error bars represent +/- standard deviation across runs.

---

> > ### Comment · Reviewer_KUYx · 2022-08-07
> > **Response + Followup**
> >
> > Thanks for your response to my questions!
> >
> > It has somewhat clarified my understanding between this work and skill chaining, but I'm still hung up on something, so I just wanted to confirm that I understand.
> >
> > As you say, the point here is that you want to generalise to any possible next task that may occur. However, much of this is as a result of the jump transitions. If you remove those, then you cannot generalise to new tracks, because the track you are currently observing is all there is, which would be the same setting as skill chaining.
> >
> > My understanding now in relation to skill chaining is as follows: in skill chaining (in particular, the work on skill graphs), you can generally only execute some finite number of next skills, because of the structure of the problem. Here though, you can execute *all* possible next skills, and so you must learn the right termination set for all of them. This is due to the jump probabilities which allows for a change of coordinates that puts the agent in the initiation set of all the skills. But now isn't this possible in the skill chaining framework too? For example, if there are appropriate state variables (e.g. agent-space variables [1], I see no reason why you could not also have a situation where all skills are executable next. This would then be the same case here, would it not? I take your statement in the response ("our approach generalizes [3] to this setting in a theoretically grounded and principled way") to mean just that. To summarise: in skill chains/graphs, perhaps not all skills are next executable, but here they are and so we need to learn one that is robust to the worst possible case.
> >
> > Please let me know if I have understood that correctly.
> >
> > [1] Konidaris, George, Ilya Scheidwasser, and Andrew G. Barto. "Transfer in reinforcement learning via shared features." (2012).

---

> > > ### Author Response · Authors · 2022-08-08
> > > **Discussion Reply**
> > >
> > > Thanks for the response. In general, ROSAC can be applied in the setting of skill chaining as well which would enable better generalization to multiple sequences of skills. Therefore, our contribution is useful in that setting as well since our algorithm tries to optimize the worst-case sequence of subtasks (as opposed to the average-case w.r.t. a distribution of tasks).
> > >
> > > The jump transitions are unknown to the learner and simply enable modeling environments with unbounded state spaces (such as arbitrarily long tracks). The formalism is a standard one adapted from hybrid automaton models. In the F1/10th example, the car observes LiDAR measurements which remains the same before and after a jump transition and hence jump transitions are only needed to model a simulator. When learning from real tracks, the jump transitions will simply be assumed to occur and we do not need to explicitly model them. Also, we can use the trained options on any track as long as we have a way to detect "end of a subtask" (this is also required in the skill chaining work) and the controller does not need to perform change-of-coordinates since it has no effect in the observation space.
> > >
> > > Also, we would like to stress that the jump transitions are available to the baselines as well. Simply introducing jump transitions and naively learning the options to complete the corresponding subtasks is insufficient to learn robust options. We achieve generalizable options by modeling the problem as a two-agent zero-sum game and solving it.

---

### Official Review · Reviewer_5XiN · 2022-07-10

**Rating:** 3
**Confidence:** 3
**Soundness:** 2 fair
**Presentation:** 3 good
**Contribution:** 2 fair

**Summary:**

The paper aims to learn robust policies for solving long-horizon tasks that are composed of sequences of subtasks. To accomplish this, the problem is framed as a zero-sum, two-player game: the agent aims to maximize its worst-case performance against an adversary who chooses the sequence of subtasks. Two algorithms are introduced to solve this multi-agent problem which demonstrate superior performance compared to a number of baselines.

**Questions:**

- Can you elaborate on the key differences between your work and the previously mentioned areas of unsupervised environment design and robust (minimax) RL?
- Can you elaborate on other real-world settings where your framework would be beneficial?
- Why does the MADDPG baseline perform so poorly, even when compared to NAIVE? I found this surprising, since even a random adversary should result in options of decent quality.

**Limitations:**

As mentioned above, I believe the authors should discuss any assumptions of control over the environment generation process (if my conclusion that this is a limitation is indeed correct). This would limit the types of tasks and settings where such a method would be applicable.

**Strengths And Weaknesses:**

**Originality**: The authors present an interesting framework for training robust options, but this framework appears closely related to previous work where an adversary designs environments which challenge the current agent (e.g. [1,2]). The main difference here seems to be that the adversary can act many times within the *same episode* to affect segments of the environment that have not yet been generated. More generally, there is a plethora of work aimed at learning robust or safe policies (not necessarily options) using a worst-case minimax objective (e.g. [3,4]). I think these areas of research should be discussed in the related work, as I'm currently not convinced that this work differs in a significant way.

**Quality**: The proposed algorithms are well-motivated, exploiting the structure of the problem to improve the learning of the adversary. They are further justified via rigorous convergence guarantees. The experiments are conducted on complex environments and demonstrate faster learning compared to a number of baselines, but I have some concerns.

- The learning curves are only reported over a small number of frames (500k) and this may not show the full picture. I'm interested in seeing whether ROSAC converges to a better solution than NAIVE/DAGGER under adversarial subtask selection. Also, I'm curious if there is any performance trade-off to training adversarially when you evaluate on uniformly random tasks.
- I think there needs to be a baseline which simply performs flat RL (preferably SAC, for a fair comparison) on the full task. If I'm understanding correctly, this is different from the NAIVE/DAGGER baselines, which instead independently try to solve each "segment" as quickly as possible.

**Clarity**: Overall the prose is very well-written but the math can be hard to follow due to the amount of notation.
The following parts could also be improved in terms of clarity:
- I think the start of Section 3 could better emphasize the properties of the problem you will be exploiting to outperform general methods for solving games (e.g. MADDPG).
- It's not clear what the NAIVE and DAGGER baselines are just from the descriptions in the experiments section.
- It's not stated what the error bars in Figs 3 and 4 are reporting.

**Significance**: Learning options that are robust and that generalize under various conditions is an important problem for RL. However, after reading the paper, I (unfortunately) remain skeptical of the impact of this particular work. First, there is an assumption that any sequence of options corresponds to a meaningful task, which is only true in a limited number of settings. Even in the racetrack example, many sequences of turns don't seem to correspond to configurations that are physically possible. The environment must also dynamically change based on the selected sequence of subtasks, requiring a great degree of control in the environment generation process. Again, I'm having trouble thinking of other settings where such a framework would be applicable.

[1]: Dennis, Michael, et al. "Emergent complexity and zero-shot transfer via unsupervised environment design." Advances in Neural Information Processing Systems 33 (2020): 13049-13061.
[2]: Gur, Izzeddin, et al. "Environment generation for zero-shot compositional reinforcement learning." Advances in Neural Information Processing Systems 34 (2021): 4157-4169.
[3]: Pinto, Lerrel, et al. "Robust adversarial reinforcement learning." International Conference on Machine Learning. PMLR, 2017.
[4]: Campero, Andres, et al. "Learning with AMIGo: Adversarially Motivated Intrinsic Goals." International Conference on Learning Representations. 2020.

---

> ### Author Response · Authors · 2022-08-02
> **Response to Reviewer 5XiN**
>
> Thanks for helpful comments and pointers to related literature. We address the main concerns below and will incorporate the discussion in the paper.
>
> __Comparison to Adversarial Environment Generation [1,2].__ These approaches model learning to perform multiple tasks as a two agent game similar to our approach but there are some key differences.
> 1. These approaches explicitly maintain and train an adversary to pick the task whereas we leverage the structure in our problem to select adversary actions using the Q functions of the agent without the need for a separate adversary policy. From the experiments, we can see that our approach outperforms the MADDPG baseline which trains a separate adversary.
> 1. They do not provide theoretical guarantees w.r.t. the worst case task since general multi-agent RL may not converge. However, in our setting, we can show convergence to the min-max policy in the finite state setting (Theorem 4.1).
> 1. In [1] a task is parameterized by a continuous parameter which the adversary tries to optimize by maximizing regret. In our setting, a task is a discrete sequence of subtasks (which might be infinite) which the adversary generates on the fly.
> 1. In [2] the tasks are similar to ones we consider, however, their experimental evaluation focuses primarily on form-filling tasks.
>
> __Comparison to min-max RL [3,4].__ While these approaches also tackle the multi-task setting, their focus is on generalizing a single policy (or a goal-conditioned policy) to perform a variety of tasks. In contrast, our focus is on performing a fixed set of subtasks in such a way that they compose well with each other and are robust options which can be used in various contexts. The way we employ min-max learning here is also novel and takes into account the compositional structure of the tasks enabling long-horizon generalization. Furthermore, general min-max RL may not always converge whereas we can give convergence guarantees in the finite state setting.
>
> __Real World Settings.__ This framework is applicable in many real-world contexts. For instance, the F1/10th example can be seen as a toy version of an autonomous driving scenario where the agent needs to learn to perform maneuvers such as turning left/right, changing lanes etc. Here, the policies for performing these maneuvers need to ensure that the car is a safe and controllable state for future maneuvers. Another interesting scenario is when a household robot has to perform multiple tasks such as cleaning, cooking etc., but needs to ensure that the policies for performing these tasks leave the house in a favorable state for future tasks – e.g., learning to cook without making the place too dirty (as it might be hard to perform the cleaning task later).
>
> __Experiments.__ We believe we conduct experiments for a large enough number of samples to conclude that we achieve better performance compared to the baselines. ROSAC achieves the best possible performance (achieves max jumps considered and completion probability of 1) within the given number of samples whereas the baselines fail to do so.
>
> __Random adversary.__ We also report the learning curves when the performance is measured against a random adversary (Figures 3a and 4a) and show that ROSAC achieves better performance in this case as well.
>
> __MADDPG.__ This baseline ignores the stage-wise structure in the problem and treats the entire game as a concurrent game where both agents take actions at every step (this is the setting that MADDPG supports). Therefore, it likely doesn’t recognize when mode switches happen and when the adversary actions are important, leading to suboptimal learning. Also, the non-stationary nature of the environment causes it to not converge, which is a common issue with general purpose multi-agent RL algorithms.
>
> __Flat RL.__ We tried the Flat RL baseline and it performed poorly as compared to NAIVE. This has also been demonstrated in prior work on compositional learning ([17] in the paper). Furthermore, we are interested in learning options rather than flat policies which is why NAIVE is a more natural baseline. We will include Flat RL results in the final version.
>
> __Arbitrary Sequences of Subtasks.__ Although we assumed that any arbitrary sequence of subtasks makes sense, this was done for simplicity of presentation. We can easily handle constraints regarding which sequences are possible by forcing the adversary to obey the constraints. For example in ROSAC, in line 13 of Algorithm 2, we can perform “argmin” over the set of subtasks that are physically possible.

---

> > ### Comment · Reviewer_5XiN · 2022-08-07
> > **Thanks for the response**
> >
> > Thank you for addressing my questions.
> >
> > - **Comparison to minmax RL**: As before, I agree that the way your approach does not require training a separate adversary is interesting and useful.
> >
> > - **Arbitrary Sequence of Subtasks**: Thanks for clarifying. I think it should also be made more clear in the paper that you can impose constraints on which subtasks are possible at any time. Does this affect any of the theoretical guarantees in the paper?
> >
> > - **Real world settings**: The examples you provided sound interesting. I think experimenting on domains with a richer set of composable options than just driving forward/turning would greatly improve the paper.
> >
> > - **Experiments**: I stand by my original comment. Reporting results over a larger number of training samples would show a great deal of useful information, including the convergent performance of the baselines, and how much (A)ROSAC improves sample efficiency. It may also lend greater credibility to the results.
> >
> > - **(new) A note on clarity/presentation**: I personally find the F1 racing example quite confusing. As I originally mentioned, it seems to require that you can dynamically alter the generation process of the environment (including potentially generating some physically impossible configurations). Assuming this level of control in the environment clearly limits the applicability, but it also seems like the framework can apply to more "realistic" settings where options are composed in a fixed environment. Unfortunately, some non-standard elements of the framework seem geared towards this F1 example, like the jump transitions (which in many cases can be hard to specify). I'm not sure that these jump transitions and the change-of-coordinates add much, and I wonder if they're really necessary.

---

> > > ### Author Response · Authors · 2022-08-08
> > > **Discussion Reply**
> > >
> > > Thanks for the response. Here are the answers to the new questions.
> > >
> > > + __Constraints on the sequence of subtasks.__ ROSAC can be applied with arbitrary constraints. Furthermore, the theoretical guarantees in the paper can be extended if the set of all possible sequences are provided in terms of a finite state automaton.
> > >
> > > + __Jump transitions.__ The jump transitions are unknown to the learner and simply enable modeling the environments with unbounded state spaces (such as arbitrarily long tracks). The formalism is a standard one adapted from hybrid automaton models. In principle, in certain environments (such as the household robot example), they can be taken to be the identity function. In the F1/10th example, the car observes LiDAR measurements which remains the same before and after a jump transition and hence jump transitions are only needed to model a simulator. When learning from real tracks, the jump transitions will simply be assumed to occur and we do not need to explicitly model them.

---

### Official Review · Reviewer_bKP4 · 2022-07-11

**Rating:** 6
**Confidence:** 5
**Soundness:** 4 excellent
**Presentation:** 4 excellent
**Contribution:** 3 good

**Summary:**

This paper mainly focuses on improving RL policy performance on long-horizon tasks, including learning efficiency and generalization. The proposed framework has a high-level objective to adversarially choose tasks to minimize the reward achieved by the low-level policies. The empirical evaluation demonstrate the method works across different domains.

**Questions:**

**Questions**

This paper is over the qualified line, and I don't have many questions. I only have two concerns :

1. Considering that the proposed method is based on the setting of multi-tasks, I was expected to find more complex tasks in the experiments. Based on the experiments, I think it is not necessary to train sub-policies for diverse sub-tasks, but a goal-conditioned RL policy is enough. What if using a goal-conditioned RL policy instead of the set of policies $\pi_1$?

2. How does ROSAC work in a more realistic task? Refer to Line.13 in Algorithm 2, what if the agent performs badly on all sub-tasks and has relatively low improvement even as the training processing. In other words, what if the initial sub-task is very difficult to learn (e,g., a 7 DoF robotic arm control)?

**Limitations:**


**Reference**

1. Benjamin Eysenbach, Ruslan Salakhutdinov, Sergey Levine. Search on the Replay Buffer: Bridging Planning and Reinforcement Learning, Arxiv, abs/1906.05253.

3. Shuang Ao, Tianyi Zhou, Guodong Long, Qinghua Lu, Liming Zhu, Jing Jiang. CO-PILOT: COllaborative Planning and reInforcement Learning On sub-Task curriculum, NeurIPS 2021.

2. R. Gieselmann, Florian T. Pokorn, Planning-Augmented Hierarchical Reinforcement Learning, IEEE Robotics and Automation Letters, 2021.

4. A. Srinivas, A. Jabri, P. Abbeel, S. Levine, and Chelsea Finn. Universal planning networks. ArXiv, abs/1804.00645.

**Strengths And Weaknesses:**

**Strengths**

1. This paper is well-written and structured. I enjoy reading it. The authors demonstrate motivations clearly and are easy to accept.

2. The theoretical explanation in this paper is sufficient and correct, good.

3. The results of the empirical evaluation are present clearly.


**Weakness**

1. The first paragraph in section 3 misses the definition of the difference between $A_1$ and $A_2$. The action of agent 2 is causing task shifting, which is one of the main differences between the proposed method, and should make it more straightforward.

2. The discussion in related work can be more substantial. It would be better to summarize the difference between ROSAC with previous work rather than only listing the related research. On the other hand, there is a line of recent work that apply planning to improve the robustness of RL policy on long-horizon tasks (e.g., [1]-[4]). It is better to include them in the discussion.

3. It would be better to include more analysis of the experiment results in section 5. It is necessary to have some in-middle results to demonstrate how two agents work together to affect the training.

---

> ### Author Response · Authors · 2022-08-02
> **Response to Reviewer bKP4**
>
> Thanks for the helpful comments and we will do our best to incorporate them in our paper. We will elaborate on the related work and also cover literature on applying planning to improve robustness.
>
> __Goal-conditioned policies.__ It is possible to share weights across subtasks, for instance, by using goal-conditioned policies. We opt to train separate policies for two reasons. (i) It is not possible to parallelize training across subtasks if we share weights and (ii) it might not be general enough to capture other scenarios where our framework can be applied. However, we feel that in certain scenarios with a large number of subtasks, goal-conditioned policies can improve sample complexity and we leave this direction for future work.
>
> __Hard to learn subtasks.__ We require that the base learning algorithm (SAC in our case) should be able to learn to complete the initial subtasks. However, the initial goal regions are usually coarse and assumed to be easy to reach (after which the goal regions get refined gradually by the learned Q functions). In more complex scenarios where that is not the case, we can use an hierarchical RL algorithm as the base algorithm and adapt our method accordingly leading to a two step hierarchy. This is an interesting direction for future work.

---

### Meta-Review · Area_Chair_sZgU · 2022-08-25

**Recommendation:** Reject
**Confidence:** Certain

**Metareview:**

This paper proposes a robust option learning algorithm that learns subtask policies to maximise the worst-case performance. The results on a 2D navigation environment and a simulated car racing environment show that the proposed algorithm achieves a better and more robust performance compared to alternative option learning approaches. Although the reviewers found the idea interesting, all of them ended up sharing several major concerns during the discussion period. First, the problem setting is not fully justified. Specifically, the assumption about the "jump transition" was not fully justified, and the problem reduces to an existing line of work on skill chaining without such an assumption. Besides, the empirical results are not convincing enough. The subtasks in the environments were so simple that it is unclear how much benefit we can get from the proposed adversarial training. In fact, the performance degradation from the random adversary to the MCTS adversary was indeed quite marginal, which also raises the same question. A good justification of the problem setting and evaluation on more complex environments would strengthen the paper. Thus, I recommend rejecting the paper.

**Award:**

No

---

### Decision · Program_Chairs · 2022-09-14

Reject